# Contactless Micro-Droplet Manipulation of Liquid Released from a Parallel Plate to an Open Region in Electrowetting-on-Dielectric Platform

**DOI:** 10.3390/mi13060898

**Published:** 2022-06-06

**Authors:** Yii-Nuoh Chang, Da-Jeng Yao

**Affiliations:** 1Institute of NanoEngineering and MicroSystems, College of Engineering, National Tsing Hua University, Hsinchu 30013, Taiwan; enochchang070@gmail.com; 2Department of Power Mechanical Engineering, College of Engineering, National Tsing Hua University, Hsinchu 30013, Taiwan

**Keywords:** EWOD, droplet manipulation, curved channel

## Abstract

In electrowetting-on-dielectric (EWOD) platform, the transfer of droplets from the EWOD boundary region (top plate and bottom plate) to the open region is challenging. The challenge is due to the resistance-like surface tension, friction from the top-plate edge, and the so-called boundary. For this purpose, we designed the top plate to minimize the friction resistance at the boundary. The experiment focused on Gibb’s formula and successfully transferred the liquid droplet between the top plate and bottom plate boundary region under a high voltage environment. The threshold voltage for the successful transportation of the droplet between the boundary is 250 V which provides strong pressure to drive the droplet.

## 1. Introduction

In recent years, more attention has been paid to many biomedical experiments due to the COVID-19 pandemic, which has boosted the requirements for high-speed performance and accelerated the development of the application of microelectromechanical systems (MEMS) in biomedical experiments [1,2,3,4,5]. Scientists are pursuing increased accuracy and stability, shorter response times, and small sample numbers. In the field of MEMS, a digital microfluid (DMF) has developed more vigorously due to nano-scale samples that enhance the surface reaction area, which reduces the experiment time. [6,7,8]. In the past two decades, this technique has shown great potential in biochemical and medical applications, including enzyme assays, cell identification, immunoassays, and DNA and protein monitoring [9,10,11,12,13]. This technique has attracted attention because it is accurate and stable and has a short response time and small sample number. DMF eliminates complicated fluidic actuation components, which draws much attention as a promising lab-on-a-chip platform with high flexibility and the capability of performing multiplex, complete cell culture [14,15,16]. In 2003, the Kim team built a digital microfluidic circuit that completed the four fundamental functions of droplet creating, transporting, cutting, and merging [17,18,19,20,21]. Electrowetting-on-dielectric (EWOD) adapts the DMF technique for the contactless manipulation of microdroplets utilizing a change in the contact angle of a droplet placed over a dielectric substrate with only electric signals, no pump and no valve [22,23]. This droplet manipulation is widely applied. Many academic researchers seek new application areas, from biomedical to optical lenses. In 2020, Anand et al. proposed a method of DNA extraction from an embryo-culture medium with an EWOD system; after the experiment, they demolded the top plate for a result on an EWOD platform [24,25]. Ying-Jhen et al. propose microfluidic patterning using an electrowetting-on-dielectric (EWOD) and liquid-dielectrophoresis (L-DEP) system [26,27]. Most EWOD bio experiments use a pipette to move a sample to the next bio experiment stage. A limitation of fluidic actuation in an EWOD system restricts fluidic manipulation between the boundaries of the top and bottom plates [28]. Removal of the top plate is typically unavoidable to collect the liquid samples, resulting in an unwished volume change, protein loss, or possible deterioration of analytes for further analysis.

At the step of removing top plates, there are two situations frequently observed, as shown in the followings. In one case, when removing the cover, the residual liquid beads leave from the lower plate and are captured on the interface of the top plate. The other situation is that a hydrophobic layer on the surface of the chip is easily contacted and fretted with the substrate during the repeated behaviors of disassembly and assembly so that the hydrophobic layer becomes damaged, which thus affects the chip lifespan. An adhesion force will be generated when the liquid beads contact the solid substrate. It has been reported that wettability is enhanced for most practical microfluid situations by the presence of micro-surface roughness. That means it is hard to pull out droplets from parallel to open space [29]. To overcome these problems, we designed a contactless procedure for the droplet manipulation to an open area for subsequent experiments without relying on any external force. From the term ‘contactless’, we point out that, by using our top plate design, the liquid droplet can be pushed out from the EWOD platform without the removal of the top plate. This design can be implemented in an EWOD system, which manipulates the droplet across the boundary, allowing the EWOD systems to integrate with more flexible design methods and applications.

A platform with customized functions can be established independent of users’ biomedical experiment system. Oliver showed evidence of the resistance of a droplet spreading at the sharp edge. The Gibbs inequality condition for the equilibrium of a drop bound by an edge has been confirmed by experiment [30]. Baratian shows that geometric constraint and electrowetting can be used to position droplets inside a wedge in a controlled way, without mechanical actuation [31], so we included the same concept in the top plate design. A 3D curve channel is proposed in this article. The curve channel can avoid the resistance contributed from the edge of the top plate when the droplet is pulled out. It is a different insight with a 3D curve design; it is common to use a wedge-shaped top plate to transport liquid beads in the parallel type EWOD system [32]. With a 3D curve channel, we can complete a continuous production in the EWOD system.

The motivation of our research was to make a top plate whereby we could retrieve the final sample (result) without detaching the top-plate structure.

## 2. Experimental Materials and Methods

### 2.1. Edge-Effect Surface-Tension Resistance

According to the previous study, it has been observed that water beads spread on a flat surface and stop at the edge of that surface. This phenomenon was explained by the droplet diffusion resistance and marginal effect discussed by Oliver in 1997 [30]. Especially in the case of a microscale, the resistance caused by the marginal effect is harshly influenced. As previously stated, the upper contact surface of the droplet must involve the edge effects that are encountered when the fluid spreads from a flat plate to an open space. The relation between the diffusion of the droplets and various phases is described with the Gibbs Equation (1) [30].

Gibbs inequality(1)$$\theta_{0}\leq \theta \leq \theta_{\mathrm{edge}}$$(2)$$\theta_{\mathrm{edge}}=\left(180{}^{\circ}-\emptyset \right)+\theta_{0}$$

Here *θ*_0_ is the static contact angle of the liquid, *θ* is the advancing contact angle of the liquid, *θ_edge_* is the angle of the liquid at the edge measured through the phase, and ∅ is the angle of the boundary of the substrate. The contact angle of the deionized (DI) water is 115° on an EWOD chip surface coated with a hydrophobic layer (CYTOP). The angle of the boundary of the chip is 90°. According to the Gibbs inequality equation, the maximum resistance angle to spread a droplet is 205°. According to the Gibbs inequality equation, an important parameter affects the resistance of the droplet spreading, and we can adjust the angle of the boundary of the substrate (∅) to decrease the resistance at the edge. It also means that for a larger contact angle of the sample on the edge, the edge has a stronger ability to stop the droplet from transfer; we, therefore, considered how to decrease the contact angle of a sample at the same time.

### 2.2. Top-Plate Design

In the previous studies, indium tin oxide (ITO) glass with a hydrophobic coating was widely used as the substrate for the EWOD systems. The ITO is supplied by Ruilong, Taiwan. However, it is frequently seen that the moving droplet on the ITO glass substrates cannot transport across the substrate edges due to their surface tension. A large degree of surface tension generally induces the droplet with a huge angle of *θ_edge_*, the angle of the liquid at the edge of the substrate surface, so that the water beads are difficult to move across the substrate edge and are separated from the chip, especially on the hydrophobic layer.

In conventional applications, the double-layered EWOD systems composed of the top plate and primary substrate are quite common in several designs. However, the repulsive hindrance contributed by hydrophobic layers to the water beads becomes incredibly large because of the increased *θ*_0_. The transport rate and efficiency of water beads in the EWOD systems tend to deteriorate. To solve these issues, we prepared a curved, hydrophobically-modified polydimethylsiloxane (PDMS) based substrate as the top plate component in cooperation with a flat PDMS based substrate below. Due to a possible application of biomedical experiments in the future, a colorless, highly transparent, biocompatible, and easily fabricated material is required; PDMS, a common and easily fabricated polymeric compound, is one of the good candidates for a substrate.

By implementing rolling the substrate, the angle of *θ_edge_* between the water bead and hydrophobic layer can be efficiently decreased. In 2016, Egunov et al. found that PDMS films have embedded stress to roll up in an organic oil environment by making a crosslinking bilayer [33]. Thus, the interaction between silicone oil and PDMS makes a deformation in a PDMS structure. We have experimentally found out that the top plate curvature and ‘silicon oil-PDMS bath time’ are directly proportional. We adapted this working mechanism of PDMS for the top-plate design. In the fabrication, the polydimethylsiloxane (Sylgard 184) mixing ratio of 184 A and B is 10:1, and it is baked at 90° for 60 min. Having accomplished this structure, we had to coat a hydrophobic layer (CYTOP) on the surface. In the final step, we must immerse the PDMS in silicone oil for 30 min.

To fabricate a curve shape as a curve channel, a specific shape of PDMS is shown in Figure 1a. We tested two shapes of structures, house and pushpin shaped, to control the curve situation.

After immersion, we obtained flat and curved interfaces on the top plate at the same time, as in Figure 1b. Both shaped PDMS increased their exposed surface area to silicone oil, increasing the reaction rate at the pin tip, but with the house-shaped design, it was easier to control the reaction rate than the other because the pushpin design had more contact area with silicone oil. The result of both shapes is shown in Figure 1c: the reaction rate is too difficult to control; the reproducibility of the pushpin-shaped plate immersing in the silicone oil is poor. As per Gibb’s formula, we found out that the house shape ∅ was 171.70°, as shown in Figure 1d, whereas the pushpin shape ∅ was 166.60°. We needed to use the minimum *θ_edge_*, which gives minimum resistance for the edge effect. Hence, we chose the house-shaped design. We thus selected the house-shaped design to make the interfacial surface have a 3D curvature.

### 2.3. EWOD Chip Design and Fabrication

We used 2019AutoCAD to mask the coplanar interdigitated EWOD design. The finger cross electrode pattern design reduces the contact angle of the droplet as well as possible. The chip consisted of 2 reservoirs electrodes, 4 transport finger cross electrodes, and an array of 4 × 5 finger cross electrodes, as shown in Figure 2a. The transporting electrodes were 0.9 mm × 0.98 mm, as in Figure 2b. The width of the electric wire was 10 μm, and the space between electrodes was 20 μm. The droplet could pass through the adjacent electrodes smoothly. Moreover, the design of interdigitated shape between the two electrodes enhanced the flawless manipulation of the droplet.

The EWOD bottom plate is made of glass and deposited by transparent Indium Tin Oxide (ITO). The process of fabrication is shown in Figure 3. The ITO-Glass wafer performed a standard clean and hot bake. Later, the ITO glass was coated with the HMDS for 5 min. Positive photoresist (AZ5214) was spin-coated, followed by exposure, development, and etching. After removing the photoresist, coat SU8-2002 acted as the dielectric layer. We selected the CYTOP coating as a hydrophobic layer.

### 2.4. Experiment Design

The EWOD platform is divided into four parts: Input, Signal Control, Output, and Monitoring and Analysis system. The input terminal is a function generator (33220A, Keysight; California, USA) and a signal amplifier (A304, A. A. Lab System Ltd., Ramat Gan, Israel) which provides signal output. The output terminal includes on board and clamp (CCNL050–47-FRC), and the output signal control is managed by PXI-6512 (National Instruments Corp, Austin, TX, USA) LabVIEW program and a microscope optical system that is used to observe droplet operations. The experiment model is shown in Figure 4a. We set up the devices on the EWOD platform. The droplet manipulation follows Figure 4b. Propylene carbonate (PC) was used as the test material. A spacer (20 μm) was set up to define the gap between the parallel plates. A silicon oil environment was used to avoid the evaporation and friction of liquid droplets. The experimental set-up was established under 10 KHz of frequency with (100–300) V. The transportation of the liquid droplet at the boundary of the two different phases was carried out by the top-plate design.

## 3. Results

### The Manipulation of Propylene Carbonate

The unique top plate design made a varying gap between the bottom plate and the top plate from 20 μm to 1.5 mm. The contact angle decreased with an increase in voltage. The 250 V at 10 kHz was the optimum level which enabled the manipulation of the droplet through open and closed regions, as shown in Figure 5.

In Figure 6, the droplet has overcome the structure boundary between the parallel and open plate.

We tried two different structures for top plate design. The major focus was on geometric design to reduce the friction effect.

A specific design of the geometrical shape of the 3D curvature surface perfectly avoided the edge effect and reduced the influence of friction to complete the mechanism of pulling out the micro-liter droplet. Here, we proposed a specific substrate design where the boundary angle of the house shape was 171.70°. This boundary angle reduced the resistance of the edge effect. Along with this, we made the ‘fillet transition’ (for smooth bending) to make the top pate surface even smoother. The threshold voltage for the successful transportation of a droplet between the boundary is 250 V, which provides strong pressure to drive the droplet. In the future, this device can be used in bio experiments such as DNA extraction, with contactless for all processes from sample extraction to PCR processing to instrument analysis. All processes can be completed on the chip; not removing the top plate reduces pollution and makes the result more stable.

## 4. Conclusions

For a micro-scale droplet, the mobility behavior of the droplet on the EWOD substrate is described with the principle of Gibb’s constraint inequality, indicating resistance to a droplet at the substrate boundary. To resolve the above-mentioned problems, a new curved surface structure of the top plate is proposed. The curve surface reduces the resistance caused by the boundary and friction effects. According to the experimental results, we can choose the house-shaped structure to conquer the resistance. For the propylene carbonate, a set of operating parameters has been provided. The manipulation of propylene carbonate from the parallel to the coplanar region has successfully been achieved. The primary design method to prove the feasibility of the droplet transfer was conducted by implementing a series of facile experiments. With this method, the residual sample loss, the chip lifespan, and the operation difficulty of the EWOD experiment have been improved. This design concept has great potential, providing EWOD users with different experimental ways to pave a new path on the EWOD platform.

## Figures and Tables

**Figure 1 micromachines-13-00898-f001:**
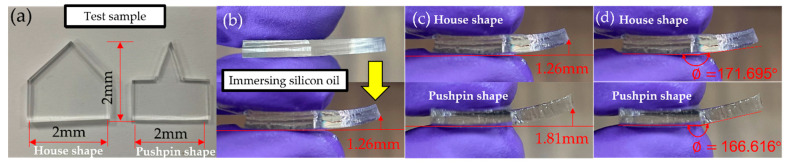
Top plate design (**a**) The left one is a house-shaped angle design; the other is the pushpin-shaped angle design. (**b**) Design of the top plate of the house-shaped structure, immersed in silicone oil for 30 min. (**c**) The result of curve top plate, the top one is house-shaped; the bottom one is pushpin-shaped PDMS. (**d**) The result of substrate angle, the top one is house-shaped; the bottom one is pushpin-shaped PDMS.

**Figure 2 micromachines-13-00898-f002:**
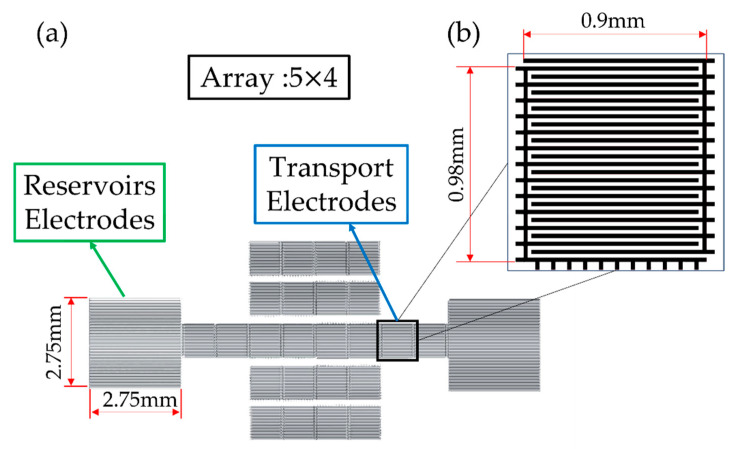
Chip design (**a**) The function of array of the mask. (**b**) The finger cross pattern design.

**Figure 3 micromachines-13-00898-f003:**
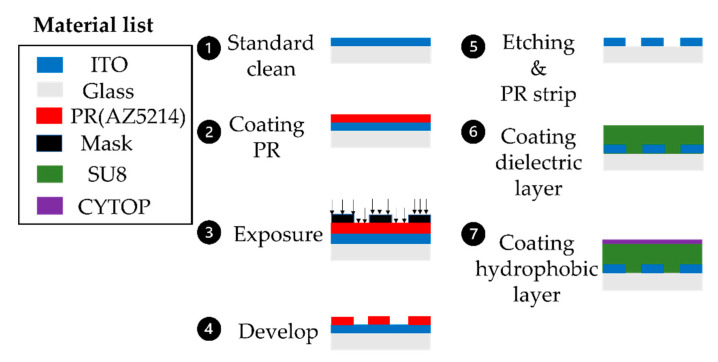
EWOD chip fabrication process.

**Figure 4 micromachines-13-00898-f004:**
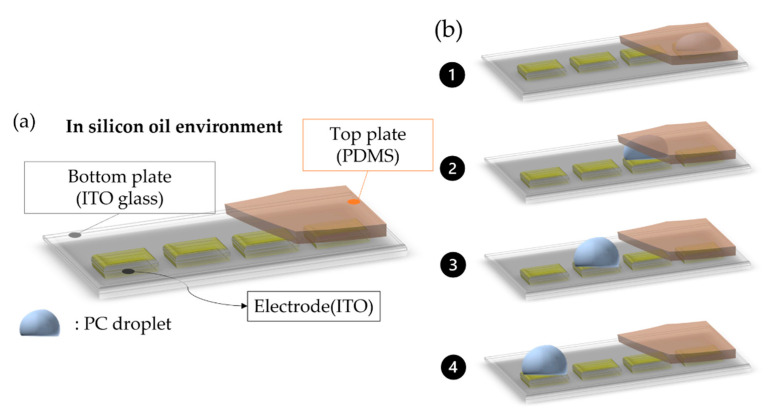
(**a**) Experiment device model. (**b**) The manipulation step 1∼4 of droplet movement from cover region to open region.

**Figure 5 micromachines-13-00898-f005:**
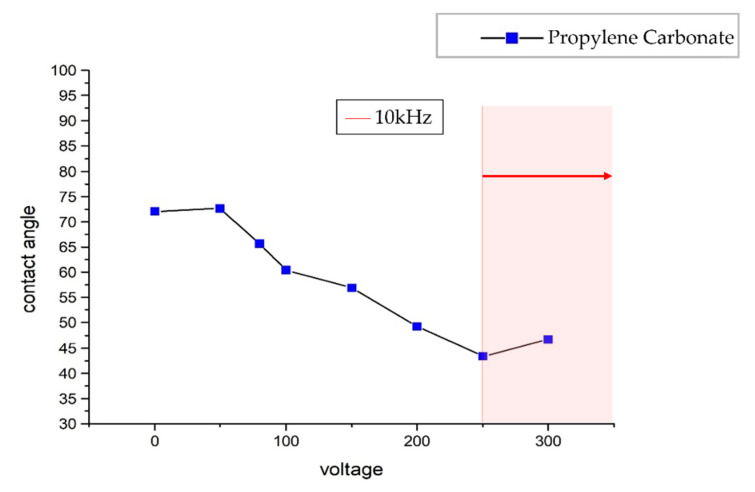
Contact angle and voltage relationship of PC in the silicon oil.

**Figure 6 micromachines-13-00898-f006:**
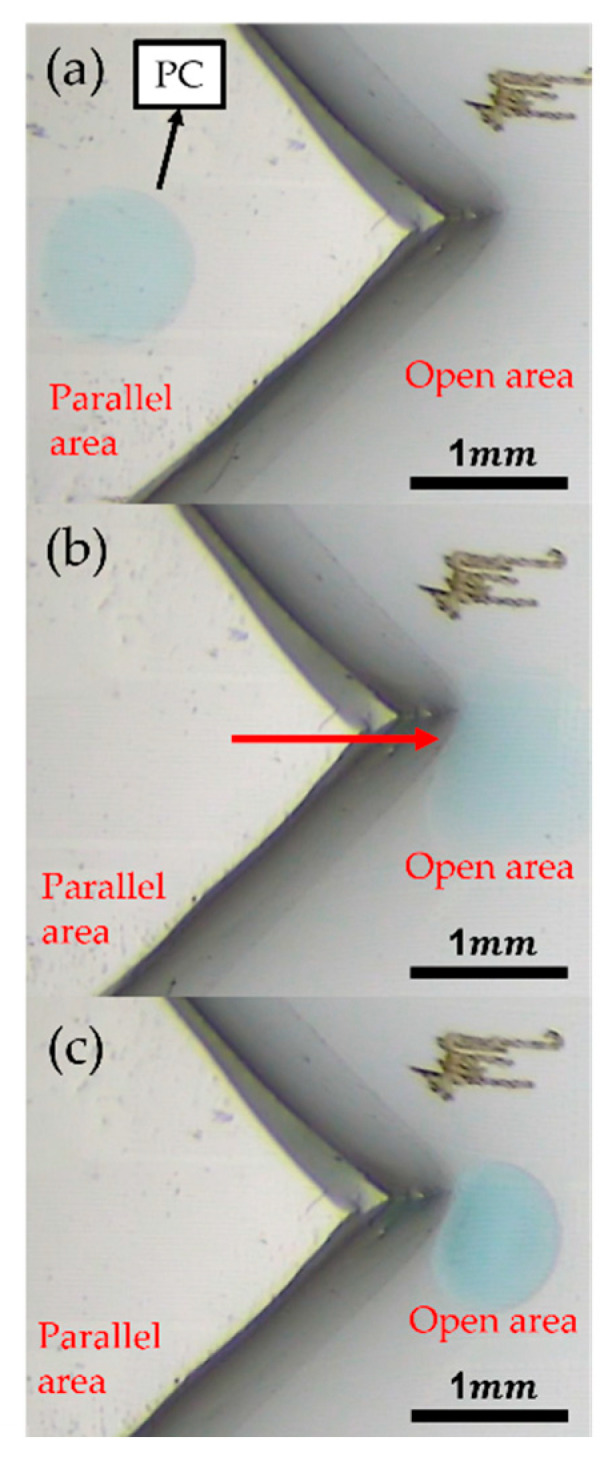
(**a**) The droplet is placed at the parallel plate. (**b**) The voltage is applied to drive the droplet. (**c**) After transportation, the phase between parallel and open plate.

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
