# Peer review of "Contactless Micro-Droplet Manipulation of Liquid Released from a Parallel Plate to an Open Region in Electrowetting-on-Dielectric Platform"

_micromachines, 2022, doi:10.3390/mi13060898_

Round 1
Reviewer 1 Report
Generally, for the EWOD droplet transport system with sandwich structure, when the droplet passes across the top-plate edge to open area, there is a great resistance to block the passage of droplets. In order to overcome the resistance, authors mainly design the house and pushpin shaped top-plate edge, and obtain the meaningful results. But there are some questions:
1) Authors have designed house and pushpin shaped top-plate edge, but only the transport process of the house shaped top-plate edge is given. How does the top-plate edge with pushpin shape?
2) Authors obtain the conclusion that 3D curvature surface can perfectly avoid the edge effect in the line 193. But How does the surface curvature affect the edge effect? There are not specific experiment results.
3) The mechanism of 3D curvature surface to reduce the resistance to droplets passing across the top-plate edge should be discussed.
Reviewer 2 Report
The authors presented an EWOD platform for droplet manipulation. A curved top plate was designed to reduce the resistance of droplet motion for the transport of droplets into an open region. This method may have advantages in reducing the loss of sample residual, improving the lifetime of the chip, and leading to an easy operation of the EWOD experiment. However, several issues need to be addressed.
1. The language needs to be improved.
2. The title and the abstract should be revised to reflect the novelty and contribution of the manuscript.
3. The Introduction should be improved to highlight the necessity of a curved top plate in EWOD droplet manipulation.
4. The authors should explain why the droplet manipulation was called “contactless”?
5. The full name of EWOD should be given when it first appears.
6. A reference needs to be cited for the Gibbs inequality.
7. What is the reason for the reduced resistance of droplet motion by using curved cover surfaces?
8. More data needs to be shown on how to control the curvature of the PDMS top plate.
9. In Fig. 4a, should the “Bottle plate” be the “Bottom plate”?
10. The scale bar is missing in Fig. 6.
Reviewer 3 Report
This paper is not well written and is not logical.
1. what is the originality? There are some papers about moving the droplets without using the plate. What is the core point of this paper?
1. EWOD in the title is not suitable for the researchers, who are not familiar with the author's field.
2. In the title, it is micro-droplet. In their paper, it is nano-droplet. However, they are different things.
3. The design of FIg.2 and FIg.4 is different.
4. In fig.3, the first step is not to clean the surface. It is coating the ITO.
5. The speed of the droplet should be investigated. and just moving is not the manipulation of the droplet.
Round 2
Reviewer 1 Report
I think the manuscript has no obvious defects and can be accepted for publication.
Reviewer 2 Report
The authors have addressed all my concerns.
Reviewer 3 Report
The authors did not deal with the questions carefully.
The paper can not meet the publishing standard of this journal.